# Evaluating the Willingness to Pay for Using Recycled Water for Irrigation

**Dimitra Lazaridou** [1,*] , **Anastasios Michailidis** [2] and **Konstantinos Mattas** [3]

1 Laboratory of Forest Economics, Faculty of Forestry and Natural Environment,
Aristotle University of Thessaloniki, 54124 Thessaloniki, Greece

2 Laboratory of Agricultural Extension and Rural Sociology, Faculty of Agriculture,
Aristotle University of Thessaloniki, 54124 Thessaloniki, Greece; tassosm@auth.gr

3 Laboratory of Agricultural Products Marketing, Agricultural Policy and Cooperatives,
Faculty of Agriculture, Aristotle University of Thessaloniki, 54124 Thessaloniki, Greece; mattas@auth.gr

* Correspondence: dimitral@for.auth.gr

**Abstract:** The present study attempts to estimate individuals' willingness to pay for recycled water irrigation, in order to enhance the water supply and ensure the continuation of irrigated agriculture in Nestos catchment. Contingent valuation method has been developed for the investigation of farmers' preferences, in monetary terms, to adopt this alternative water source for irrigation purposes. The applied method is regularly followed in the framework of environmental valuation. The results of the survey are based on data collected from questionnaires, which were answered by respondents at a river basin scale. In a representative sample of 302 farmers, we find that 64.2% of them expressed a positive stance towards using recycled water, a fact that results in lower environmental impacts. However, findings indicate that participants are willing to pay a significantly less amount of money than they already pay, for freshwater. Additionally, the analysis demonstrates that the use of recycled water in agriculture is more acceptable to respondents who are aware of its environmental benefits. Therefore, the provision of complete information on the welfare of using recycled water for irrigation to farmers may lead to greater adoption intention and a greater environmental benefit.

**Keywords:** willingness to pay; contingent valuation method; recycled water; environmental benefits; public acceptance

## 1. Introduction

In many countries worldwide, fresh water scarcity has already emerged. The agricultural sector exacerbates this problem, as it is the main water user and accounts for 90% of water demand per year [1]. In Greece, a correspondingly high percentage is recorded and reaches of 83% [2]. At the same time, irrigation is recognized as the main contributor to water degradation [3]. In order to address this issue, alternative water resources have been proposed, globally. Among them, the reuse of urban waste in agricultural processes presents a growing adoption. As the water purification technologies advance, municipal wastewater could be reclaimed and reused for landscape and crop irrigation purposes. This new resource provides a viable opportunity to complement water supplies in a safe and sustainable way [4]. Besides, it guarantees a high level of supply reliability given the capacity to provide a constant volume of water. Wastewater reuse for non-potable purposes has environmental, social, economic impacts. Benefits may arise from its use in agriculture are summarized in nutrient capture and fertilizer savings, improvement of the soil microorganism activities, enhancement of soil health conditions, higher crop yields, salinity decrease, avoiding of freshwater pumping and energy savings which entail carbon footprint reduction [5–7]. It is claimed that wastewater reuse

can contribute towards climate change mitigation, given that it could mitigate the water footprint of food production on the environment [5]. Meanwhile, recycled wastewater may be used to rejuvenate the previous characteristics of the water bodies' ecological status [8]. In addition, this new resource creates supply-side benefits enhancing the local economy, as well as becoming a satisfactory factor to guarantee socio-economic stability [9].

On the other hand, there exist studies that support opposing results, focusing on potential negative impacts. According to them, reclaimed wastewater involves a certain amount of environmental risks and human health risks, via consumption or exposure to pathogenic microorganisms. It maybe contains dissolved solids, heavy metals, harmful organic chemicals pathogens and other substances which may increase the salinity and sodium content in soil and cause damage to ecosystems, crops or human beings [10,11].

As deduced, public involvement is critical to the implementation of water reuse programs, given that their success depends on acceptance and support from the potential users [7,12]. There is a need to explore the role that farmers' perceptions played in the adoption of recycled water. This is necessary not just in terms of identification of factors influence their attitudes, but also in investigating their willingness to use and to pay for recycled wastewater irrigation.

The literature review reveals an amount of surveys that examined farmers' behaviour about the option of wastewater reuse, in different countries. Specifically, farmers' attitude towards this alternative source investigated in Bangladesh [13], in Iran [14], in Jordan [15], in Israel [16], as well as in Greece [17]. There exists also a body of literature that raises some doubts and reveals a reluctance in acceptance of recycled water in agriculture [2,18,19]. However, only a few papers evaluate the challenges of farmers' willingness to pay for recycled water [20–23].

Therefore, the main objective of this study is to analyze the socio-economic aspects around the concept of wastewater reuse and to estimate the value farmers attach to safe use of recycled water for irrigation purposes. Since it is investigated the willingness to pay for a nonmarket good, recycled water use, this study applies the contingent valuation method (CVM).

## 2. Research Method

### 2.1. Research Area

Nestos River flows through a north area of Greece (prefectures of Drama, Xanthi and Kavala), it is depicted in Figure 1. The main channel of the river is approximately 235 km. At the estuary of the river an extensive deltaic plain is formed, where agriculture areas occupied a great extent. The exploitation of river's flow for agriculture purposes contributes decisively to the expansion of irrigated agriculture in the region. Thus, a relatively high percentage of area's population is partly or fully employed in agriculture sector, while agriculture plays a crucial role in local financial development and in the total Gross Domestic Product (GDP) produced there [24].

In the greater part of the cultivated land, irrigation water distributed through three Local Organizations for Land Reclamation and a smaller part of the region is irrigated by numerous drillings and collective irrigation networks. With regard to the irrigation water prices, they range between 90 €/ha and 135 €/ha, depending on the irrigation water provider. At the same time, the cost for farmers who use drillings for irrigation is comparatively much greater [25]. According to respondents' statements it reaches 209 €/ha. So, the mean price of the current charge, that participants stated, calculated 16.47 €/ha.

As would be expected, the extensive use of agrochemicals and drills for irrigation purposes have a knock-on effect on the water quality and quantity in the area. Particularly, the overexploitation of groundwater in the coastal zone has led to depleted aquifers [26]. Meanwhile, increasing water demands for agricultural activity combined with climate change are projected to deteriorate water resources quantity [27].

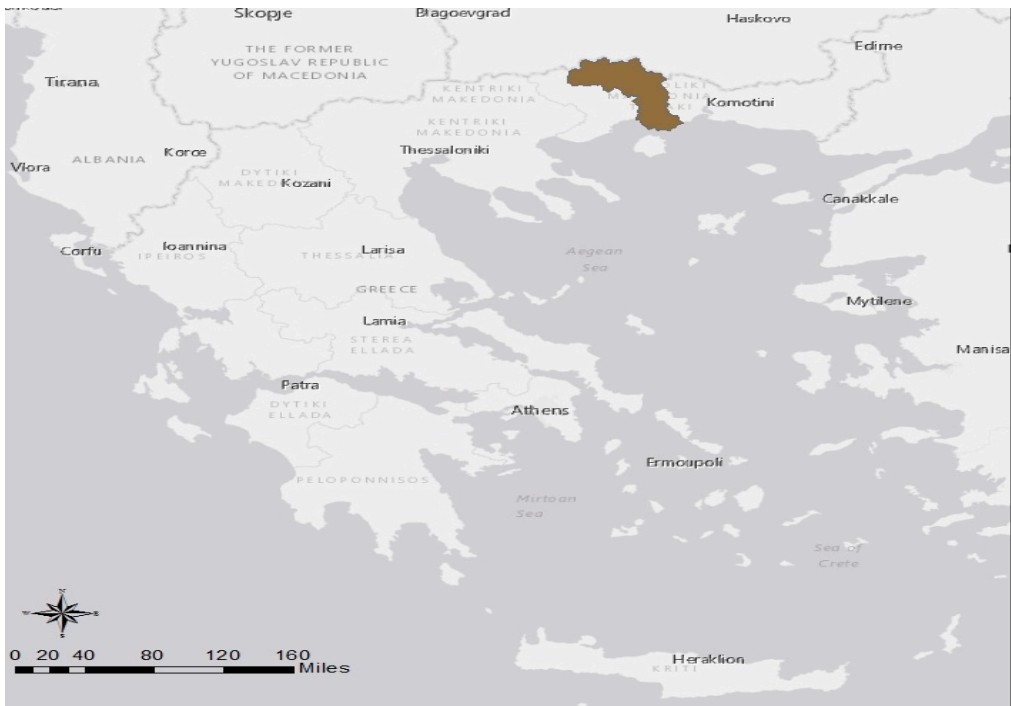

**Figure 1.** Geographical location of Nestos watershed, Greek part (modified in ArcGIS 10.1).

## 2.2. Specific Method

This survey aims at putting a price on an environmental good that is not traded in the market, therefore stated preference methods are used to elicit this price. Among the other methods the Contingent Valuation Method (CVM) was selected to address the aim of this survey, given that environmental economists have traditionally applied this approach for water quality valuation [28]. The method based on asking to a representative sample of the population, in a hypothetical market is presented, how much they are willing to pay for the improvement of a public good related to their wellbeing.

As in many other CV studies, the questionnaire contained a number of validities checking questions that help to interpret the WTP estimate. These involved socio demographic, attitudinal and behavioural questions, questions regarding agriculture and irrigation practices, views towards the environment and questions related to awareness about the water degradation problems. The survey also included the CV hypothetical question, which constitutes the most crucial part of the method. At that stage respondents are presented with general information concerning the good, namely the recycled water, they would be willing to pay for use it. The disclosure of the hypothetical scenario is followed by the valuation question. There respondents were asked to answer the question if they are willing to pay for recycled water use, in case the program will implement in their region. In our CV survey, the WTP question was elicited by an open-ended format, where individuals are directly asked to state their maximum WTP for the good. After the quote of the scenario, participants who rejected it asked to state the main reasons for their refusal, through a final debriefing question. Debriefing question designed to obtain more information on farmers' motivations and attitudes.

During the field study face-to-face CV interviews were administered. According to [29], in person interviews facilitated the researchers to engage with the farmers and remove their suspicion and doubts regarding the survey. The survey was performed from November 2016 to April 2017, among 302 farmers (out of a total of 2551) in the Nestos catchment. The sample size has been estimated in order to fulfill the requirements of Slovin's formula for finite population, is given from the following equation [30,31]

$$n = \frac{N}{(1 + Ne^2)} \tag{1}$$

where $n$ defines sample size; $N$ is the population and $e$ defines the accepted margin of error (5%). A pre–test procedure was conducted in 20 farmers to identify the reliability and validity of the questionnaire.

The multiple linear regression analysis was performed to evaluate the mean WTP for recycled water use. Multiple regression has been revealed as the appropriate technique of choice for predicting a price of a dependent variable, based on the values of a set of independent variables. The general form of the regression model is given from the following linear equation:

$$y = a + b_1x_1 + b_2x_2 + b_3x_3 + \ldots + e \tag{2}$$

where $y$ defines the dependent variable; $a$, $b_1$, $b_2$, $b_3$ are the partial regression coefficients; $x_1$, $x_2$, $x_3$ the explanatory variables; while $e$ is the error on the multiple linear model.

The ultimate goal of fitting a statistical model to CV responses is to derive a measure of the WTP distribution and to estimate the welfare change to society, due to a proposed scenario [32]. The multiple linear regression model was created using the Statistical Package for Social Sciences (SPSS 25.0) software. Throughout the procedure of model development, several linear regression analyses were performed.

## 3. Research Findings

As mentioned above, this study yielded 302 usable responses from farmers who work in the Nestos plain. The self-reported of socioeconomic and demographic characteristics of the sample are summarized in Table 1.

**Table 1.** Socioeconomic and demographic characteristics of respondents.

| Variable | Value | Frequency | Percentage (%) |
|----------|-------|-----------|----------------|
| Age | 20–29 | 7 | 2.3 |
| | 30–39 | 47 | 15.6 |
| | 40–49 | 58 | 19.2 |
| | 50–59 | 78 | 25.8 |
| | 60–69 | 80 | 26.5 |
| | 70–79 | 29 | 9.6 |
| | >80 | 3 | 1.0 |
| Gender | Female | 43 | 14.2 |
| | Male | 259 | 85.8 |
| Education level | Illiterate | 11 | 3.6 |
| | Primary and Secondary school | 187 | 61.9 |
| | High school | 52 | 17.2 |
| | Associate degree | 19 | 6.3 |
| | University | 33 | 11.0 |
| Mean monthly Income | 1.307 € | | |

Attention was paid to the gender distribution in the conducting of the questionnaires. However, 85.8% of the questionnaires were filled out by male respondents and 14.2% by female. An explanation for this is the fact that, traditionally, male employment in farming numerically overcomes the women's employment. Of all respondents, the average age was 53.4 years. The education levels of the interviewees range from illiterate to University graduate. The assessment of the educational status of the respondents showed that those have received primary and secondary education are in the first position with 61.9%, followed by high school graduates at 17.2%. About 11% had graduated with a University degree.

At the same time, the analysis of the findings shows that only 34.4% of the farmers are adequately knowledgeable about the environmental benefits of wastewater reuse. It is also observed that recycled water to have lower acceptance than the freshwater. 91% of farmers declare that would prefer freshwater for crops irrigation, if they have the possibility to select between the two alternative choices.

The participants were presented with general information concerning the proposed scenario (project). The research found that after exposure to more information, through the scenario, 64.2% of farmers were willing to pay for using recycled water in irrigation agriculture. The refusal rate, in the hypothetical proposal, reaches the percentage of 35.7%. Although high, it is within an acceptable range and finally included in the econometric analysis. It is believed that the exclusion of refusal responses from the full sample may alter its structure and make it less representative of the population [33]. This fact might induce a sample selectivity bias [34,35]. As described above, the mean WTP estimate obtained through an econometric model. The independence variables compose the model are cited in Table 2.

**Table 2.** Parameter coefficients compose regression model (n = 302).

| Variables | *B* | Coefficients Std. Error | Sig. |
|---|---|---|---|
| IrrigWatPric | 0.160 | 0.030 | 0.001 |
| KnowledgRecWat | 2.313 | 0.679 | 0.001 |
| Education | 0.058 | 0.130 | 0.003 |
| Age | −0.052 | 0.033 | 0.001 |
| Constant | −1.444 | | |

Previous study has already noted that stakeholders' perception of recycled water is affected by their social-economic backgrounds [7]. Explanatory variables and coefficients with the corresponding signs indicate the determinants of farmers' willingness to pay. Particularly, farmers with higher educational level have a higher probability to pay for reclaimed water reuse. This link between education and acceptance of recycled water is highlighted by the literature [23,36]. It is also predicted by the model that greater consent to water reuse schemes would be from younger farmer. The abovementioned results are consistent with empirical findings [37,38]. In addition, results imply a relationship between knowledge of recycled water and its acceptance. This finding is concluded by some other studies, which show that general knowledge about water treatment processes is significantly related to greater acceptance of reuse of water [36,39].

Finally, it is emerged from the results that farmers' willingness to pay for the proposed program also explained by the current price they already pay for irrigation. Consequently, participants who pay more for irrigation water are more willing to pay for recycled wastewater irrigation. This finding may be explained by participants' expectation to irrigate at a lower cost through the reclaimed water. Maybe interviewees expect to pay less for using recycled water, due to the perceptions of lower quality and the restrictions on people's use it. The mean price farmers are willing to pay for using recycled water in irrigation agriculture is 20.54 €/ha, as it was calculated from the function 2. In case of scenario rejection, farmers were asked to declare the reasons for their refusal to adopt the alternative water resource. So, their attitudes towards water reuse are in turn determined by some closed-ended debriefing questions.

The results presented in Table 3 show that a significant percentage of participants (32.7%) do not recognize the need to adopt alternative water resources, in the catchment of Nestos river. 17.2% of the respondents declared unwillingness to pay for recycled water irrigation, because of their distrust in authorities to manage risk may arise from inadequate water reclaimed processes. Previous results reported in the literature substantiate that, distrust of authorities emerges as a common concern related to acceptance of recycled water [40]. Concerns about the potential risks to irrigated crops is another preferred option, with nearly 13% of the farmers selecting this choice. Furthermore, recycled water was judged as disgusting by 11.8% of farmers. There are authors who mention [41] that the emotion of disgust, or the so-called yuck factor, motivate resistance or rejection attitudes from farmers. Disgust

is considered as a critical factor on the acceptance of recycled water in the broader literature [42,43]. Besides, there is a percentage of farmers (9%) opposed to recycled water due to concerns for degradation of commercial value of their cultivated products. Finally, environmental-related risk (the quality degradation of the existing surface-water and groundwater) emerges as a major concern of participants, which leads them to withdraw or turn away from recycled water.

**Table 3.** Reasons for those who rejected the recycled wastewater irrigation.

| Reasons for Rejection of the Proposal | Rejection Rate (%) |
| --- | --- |
| No need for recycled water use | 32.7 |
| Lack of trust in authorities to provide safe recycled water | 17.2 |
| Distrust in safety of recycled water quality | 14.5 |
| Worry for negative impacts on irrigated crops | 12.7 |
| Disgust to the recycled water term | 11.8 |
| Concern for degradation of products commercial value | 9.0 |
| Potential pollution risk of water resources quality | 1.8 |

## 4. Conclusions

The study contributes to the literature by applying a stated preference method to estimate farmers' WTP for water reuse in irrigation. As it arises from the outcomes, participants expressed a positive stance towards using recycled water, given that 64.2% of them are willing to pay for its use. However, their knowledge about its potential application in agriculture has proven to be restricted and hence it needs to be improved. Besides, respondents' perceptions on reclaimed water reuse are affected, mainly, by their social backgrounds. Moreover, the investigation of reasons for its rejection shows that distrust in its safe use and lack of a strong need to adopt alternative water resources are the driving forces that determine their attitude.

Estimating the willingness to pay for recycled water concludes that farmers attribute a lower value to recycled water comparatively to fresh water. They are willing to pay annually on average 20.54 €/ha of recycled water, which corresponds 12.7% of the average price they stated that already pay for fresh water. A possible explanation for this low estimated rate is the fact that there are not serious problems of water deficiency in the catchment.

In any case, evaluating the willingness to pay for irrigation water can supply beneficial information to policy decision-makers [44]. Such evaluations can provide decisions with valuable information that might be used in the implementation of formulating strategy in water management. In addition, policies made informed regarding the factors leading to negative attitude for recycled wastewater irrigation, so those negative aspects can be managed successfully in order to encourage integrated water resources management.

**Author Contributions:** Writing—original draft, D.L. and A.M.; Writing—review & editing, A.M. and K.M.

**Funding:** This research received no external funding.

**Acknowledgments:** We would like to thank respondents who patiently participated to the research, as well as the reviewers for their helpful comments.

**Conflicts of Interest:** The authors declare no conflict of interest.

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
