# Peer review of "Evaluating the Willingness to Pay for Using Recycled Water for Irrigation"

_sustainability, doi:10.3390/su11195220_

Round 1

Reviewer 1 Report

Whereas it uses interesting methods, the econometric analysis need to be developed further.

It would be useful to add more statistical results, if available.

As author states 35.77% of respondents are not willing to pay for using recycled water. However, these responses included in the econometric analysis. Is it a common way to treat zero responses? Maybe it needs more explanation.

Author states that “the sample of 302 farmers is representative”. How it turns out??

Checking for some language and grammar mistakes.

Author Response

Whereas it uses interesting methods, the econometric analysis need to be developed further.

It would be useful to add more statistical results, if available. Unfortunately, there are no more statistical results available.

As author states 35.77% of respondents are not willing to pay for using recycled water. However, these responses included in the econometric analysis. Is it a common way to treat zero responses? Maybe it needs more explanation. It has been explained.

Author states that “the sample of 302 farmers is representative”. How it turns out??  It has been explained.

Checking for some language and grammar mistakes. Text has been checked for language and grammar mistakes. It has been checked.

Reviewer 2 Report

In section 2, Research Method, give price paid for irrigation for Nestos River farmers. It is mentioned later as 160.47/ha, which is higher than the range shown here, 90-135.

In equation (1), use the same 'B' (beta?) variable that is given in Table 2 so that the reader sees that these are the same.

You often give percent results in 4 significant figures. Is this realistic? I think that three sig figs is plenty, e.g., "64.2% of farmers"

line 139 ff .    How did you know that only 34% of the farmers surveyed were "adequately knowledgeable about the environmental benefits"? Was a pre-survey given to determine this? Please explain as this would be of great benefit to the reader. 

line 144 ff       "...after exposure...participants were more supportive of adoption." You give a number of 64.2% were WTP. Is this after exposure? What was the number before exposure? This will be of great interest to the reader.

line 169 .  The mean WTP is calculated as 2.054. But then in the discussion you say that it is 20.54/ha for recycled water. Can you explain to the reader how / why those two numbers are different, please?

Author Response

Reviewer 2

In section 2, Research Method, give price paid for irrigation for Nestos River farmers. It is mentioned later as 160.47/ha, which is higher than the range shown here, 90-135. It has been clarified in the text

In equation (1), use the same 'B' (beta?) variable that is given in Table 2 so that the reader sees that these are the same. Yes it is Beta. These are the regression coefficients that used in the regression model. B has been corrected.

You often give percent results in 4 significant figures. Is this realistic? I think that three sig figs is plenty, e.g., "64.2% of farmers"  It has been corrected.

line 139 ff .    How did you know that only 34% of the farmers surveyed were "adequately knowledgeable about the environmental benefits"? Was a pre-survey given to determine this? Please explain as this would be of great benefit to the reader. The results arose from the current survey. We know that because respondents were asked about this.

line 144 ff       "...after exposure...participants were more supportive of adoption." You give a number of 64.2% were WTP. Is this after exposure? What was the number before exposure? This will be of great interest to the reader. There was no corresponding question before the presentation of the scenario. The sentence has been clarified in the text

line 169 .  The mean WTP is calculated as 2.054. But then in the discussion you say that it is 20.54/ha for recycled water. Can you explain to the reader how / why those two numbers are different, please? It has been corrected.

Reviewer 3 Report

Please work on the following.

Make sure you are using the same font and size, as that does not appear in the printout. There are several sentences that need attention. I encourage you to make the paper edited. The background is not sound. I still do not find a solid answer why the study is done.  The survey is very limited and the authors need to acknowledge that. I have never read a paper with so few questions. What about the size of the firm? About the economic condition of the respondent? Abundance of water supply? There is a lot to address. (Page 2, line 78-82) According to my knowledge, drip irrigation is a system of irrigation. What that has to do with price of water? Equation 1, use equation editor.

Based on the data, you have limited score to improve. But, I suggest you at least explain the research ground, survey and the result sufficiently. For example, at least mention the survey questions. That is, explain the first two variables as mentioned on table 2 (IrrigWatPric, KnowledgeRecWater).

Author Response

Reviewer 3

Make sure you are using the same font and size, as that does not appear in the printout. Thank you. It has been corrected.

I have never read a paper with so few questions. What about the size of the firm? There is no firm. As it is mentioned, the population is 2,551.

About the economic condition of the respondent? Mean monthly household income  has been added.

Abundance of water supply? There is a lot to address. A related paragraph has been added.

(Page 2, line 78-82) According to my knowledge, drip irrigation is a system of irrigation. What that has to do with price of water? The sentence has been clarified in the text

Equation 1, use equation editor. It has been corrected.

Based on the data, you have limited score to improve. But, I suggest you at least explain the research ground, survey and the result sufficiently. For example, at least mention the survey questions. That is, explain the first two variables as mentioned on table 2 (IrrigWatPric, KnowledgeRecWater). These variables are explained in the text.